# Gut Microbiome and Small RNA Integrative-Omic Perspective of Meconium and Milk-FED Infant Stool Samples

**DOI:** 10.3390/ijms24098069

**Published:** 2023-04-29

**Authors:** Polina Kazakova, Nerea Abasolo, Sara Martinez de Cripan, Emili Marquès, Adrià Cereto-Massagué, Lorena Garcia, Núria Canela, Ramón Tormo, Helena Torrell

**Affiliations:** 1Eurecat, Centre Tecnològic de Catalunya, Centre for Omic Sciences (COS), Joint Unit Universitat Rovira i Virgili-EURECAT, Unique Scientific and Technical Infrastructures (ICTS), 43204 Reus, Spain; 2Clínica Terres de l’Ebre, 43500 Tortosa, Spain; 3ESPGHAN, European Society for Paediatric Gastroenterology, Hepatology and Nutrition, 1201 Geneva, Switzerland; 4Gastroenterology and Nutrition Pediatric Center, 08006 Barcelona, Spain

**Keywords:** bacteriome, virome, smallRNome, gut microbiota, meconium, holobiont, multiomics, metagenomics, breast-fed, formula-fed

## Abstract

The human gut microbiome plays an important role in health, and its initial development is conditioned by many factors, such as feeding. It has also been claimed that this colonization is guided by bacterial populations, the dynamic virome, and transkingdom interactions between host and microbial cells, partially mediated by epigenetic signaling. In this article, we characterized the bacteriome, virome, and smallRNome and their interaction in the meconium and stool samples from infants. Bacterial and viral DNA and RNA were extracted from the meconium and stool samples of 2- to 4-month-old milk-fed infants. The bacteriome, DNA and RNA virome, and smallRNome were assessed using 16S rRNA V4 sequencing, viral enrichment sequencing, and small RNA sequencing protocols, respectively. Data pathway analysis and integration were performed using the R package mixOmics. Our findings showed that the bacteriome differed among the three groups, while the virome and smallRNome presented significant differences, mainly between the meconium and stool of milk-fed infants. The gut environment is rapidly acquired after birth, and it is highly adaptable due to the interaction of environmental factors. Additionally, transkingdom interactions between viruses and bacteria can influence host and smallRNome profiles. However, virome characterization has several protocol limitations that must be considered.

## 1. Introduction

The human gut harbors more than 100 trillion different microorganisms (mainly bacteria, but also viruses, protozoa, archaea, and fungi) that are in a symbiotic relationship with the host. The gut microbiota plays a crucial role in human health, and there is crosstalk between the intestinal microbiota and immune development, metabolism, neurogenesis, gastrointestinal integrity, and many other systems across the lifespan, beginning during fetal development [1,2]. Thus, the role of the microbiota in numerous intestinal and extraintestinal diseases has become increasingly apparent [3,4], and more research is now being performed on how alterations in the early gut microbiota could influence child and adult health [5].

The acquisition and development of the gut microbiota in infancy is generally believed to begin at birth, with the mode of delivery being the first contact with microbiota (vaginal population for vaginal delivery and skin population for cesarean delivery) [6]. Although the womb has long been assumed to be sterile, some evidence of nonpathogenic bacteria in the placenta, amniotic fluid, and fetal gut has questioned this concept [7,8]. Regardless of this debate’s outcome, microbes from maternal and environmental sources rapidly and densely colonize the neonate at birth. The development and maturation of the gut microbiota are dynamic and nonrandom processes, where positive and negative interactions between significant bacterial taxa start within a few hours after delivery [9,10,11]. Following birth, another factor that determines early colonization is the type of milk feeding: breast-fed or formula-fed. It has been suggested that breastfeeding can have a protective effect on illness development due to its repercussions on gut microbiota composition, for example, in celiac disease [12]. The mother’s age, length of gestation (i.e., full-term versus preterm), smoking habits, and body mass index are also important factors to be considered. After the first 6 months of life, the gastrointestinal tract slowly acquires a more complex bacterial community, which substantially increases its diversity when solid food is introduced. Microbiota diversity expands during the first two years of life. At this moment, the community converges toward an adult-like state, and it remains stable throughout the years until old age, the moment when microbiota diversity starts decreasing [13].

Although the microbiota is mainly composed of bacteria and viruses, most published microbiome studies have focused on the bacterial community, as they clearly dominate the microbiome, and techniques used for bacterial community studies are more developed and standardized than those for the viral population (namely, the virome) [14]. However, the gut virome is highly correlated with the intestinal bacterial population, and its main composition corresponds to the Caudovirales order [15,16], formed mostly by bacteriophages. Bacteriophages are viruses that infect bacteria, and thus have various mechanisms that could control the density, diversity, and network interactions inside gut-associated bacterial communities, such as lysogeny and gene transfer [17,18]. The gut virome is also acquired from an early age, and similar to the bacteriome, many factors influence its shaping (e.g., type of feeding) [19].

Although the external environment plays an important role in shaping the gut microbiome community, the host itself can modulate the microbial ecosystem through different mechanisms, such as epigenetic factors, including small RNAs.

Small RNAs consist of different types of regulatory small transcripts, with microRNAs (miRNAs) being the best known and most studied class, and have been described as key regulators in multiple cellular functions. miRNAs are a family of very stable small noncoding RNAs containing approximately 20 nucleotides that regulate gene expression [20], and changes in their expression and function can be associated with numerous diseases. Recent studies have suggested bidirectional interactions between host cells and the gut microbiota via miRNAs that participate in shaping the gut microbiota after they are secreted from intestinal epithelial cells. Likewise, host miRNA expression can be influenced by the microbiota through microbe-derived metabolites that might potentially influence the host physiology [21,22,23,24].

To gain a better understanding of how the gut microbial composition is shaped in early life stages, we conducted a fecal holo-omic study in newborn infants (meconium samples were obtained) and in exclusively milk-fed infants, either with human milk (breast-fed group) or formula milk (formula-fed group). This study included the analysis of fecal bacterial and viral populations, as well as the identification of host small RNA signaling in feces.

## 2. Results

### 2.1. Bacteriome Analysis

The microbiota population at the phylum level showed differences among the three groups. Proteobacteria was the predominant phylum in meconium samples (41.6%) and in the formula-fed group (46.4%), mainly due to the *Enterobacteriaceae* family, which was less prevalent in the breast-fed group (27.8%). The distribution of the Firmicutes phylum was similar between the three experimental groups (30–33%), and the fundamental difference in the meconium group compared to the two other groups was the higher presence of the Bacteroidetes and Actinobacteria phyla. However, none of these differences were statistically significant. In contrast, the Verrucomicrobiota phylum showed that the *Akkermansia* genus was significantly different in the formula-fed group, whose abundance was lower in comparison to the other two groups (*p* < 0.05).

Nevertheless, α diversity measured using the Shannon index remained unchanged between groups (Figure 1B). However, when β diversity (between-sample diversity) was measured by unweighted UniFrac and compared using ANOSIM and PERMANOVA analysis, both tests showed significant segregation of the bacterial composition across different categories (Figure 1A), with the breast-fed group showing the most differentiated cluster (ANOSIM *R* = 0.07, *p* = 0.03; PERMANOVA pseudo-*F* = 2.11, *p* = 0.02).

Afterwards, split statistical analysis was performed to determine the differences between meconium and stool samples from milk-fed infants to assess the effect of the feeding type.

#### 2.1.1. Taxonomic Differences in the Gut Bacteriome of Meconium, Formula-Fed, and Breast-Fed Infants

The statistical comparisons of the taxa among the meconium group versus the other two groups showed that the *Prevotellaceae* family (*p* < 0.001), specifically *Prevotella* (*p* < 0.001), *Paraprevotella* (*p* < 0.001), and *Alloprevotella* (*p* < 0.01) genera, were higher in the meconium group. There was also a higher presence of *Treponema* (*p* < 0.01) and Rikenellaceae (*p* < 0.001) genera in this group. On the other hand, *Enterococcus* (*p* < 0.01), *Epulopiscium* (*p* < 0.01), and *Lactobacillus* (*p* < 0.05) genera were lower in the meconium group compared to the other two.

Separately, the *Enterobacteriaceae* family, represented by *Citrobacter* (*p* < 0.01), *Enterobacter* (*p* < 0.01), and *Klebsiella* (*p* < 0.05) genera, was less abundant in the meconium samples compared to the formula-fed group, in contrast to *Akkermansia* (*p* < 0.05) and *Muribacullaceae* (*p* < 0.05) genera, which are more abundant in the meconium samples.

When comparing the meconium and breast-fed groups, the *Veillonella* genera (*p* < 0.05) had a lower presence in meconium samples.

#### 2.1.2. Influence of Feed Type on Microbiota

When comparing the two types of feeding, the formula-fed group presented higher levels of *Prevotellaceae* and *Enterobacteriaceae* families and a lower abundance of *Bacteriodaceae* (Figure 2), and these differences were maintained at the genus level, as already mentioned above. Additionally, at the genus level, the formula-fed group showed higher levels of *Micrococcus*, *Dorea*, and *Lactococcus* (*p* < 0.05). In contrast, the *Lactobacillae* family was more abundant in the breast-fed group (*p* < 0.05), while the genera *Parasutterella*, *Butyricimonas*, *Desulfovibrio,* and *Acinetobacter* (*p* < 0.05) were more abundant.

### 2.2. Virome Analysis

Since viral enrichment was performed from stool samples, DNA and RNA virus analyses were performed separately and interrogated at the species level. After sequencing, only a small percentage of reads could be assigned to viral taxonomy (19.14% for DNA viruses and 1.86% for RNA viruses).

#### 2.2.1. Differences in DNA Virome, but Unchanged RNA Virome

The sequences obtained from the DNA virome were assigned to 495 different species, predominantly of the order Caudovirales (phages). For RNA viruses, 54 different species were identified: 53.7% of species belonged to bacteriophages, 31.5% to animal viruses, 7.4% to cloning vectors, 3.7% to phytoviruses, and the remaining 3.7% were crAssphages or undetermined.

There was no difference between α-diversity (measured using the Shannon index) values among groups in either DNA viruses or RNA viruses. However, when β-diversity was measured by Bray–Curtis and compared using ANOSIM and PERMANOVA analysis, both tests showed significant segregation of the DNA viral composition across different categories (Figure 3A), with the breast-fed group showing the least differentiated cluster (ANOSIM *R* = 0.29, *p* = 0.01; PERMANOVA pseudo-*F* = 2.47, *p* = 0.01). Any difference was observed when comparing RNA viral populations (Figure 3B).

#### 2.2.2. Presence of Exclusive Bacteriophages in Milk-Fed Groups

*Acinetobacter* phages, *Bifidobacterium* phages, *Lactococcus* phages, *Leuconostoc* phages, *Mycobacterium* phages, *Stenotrophomonas* phages, and *Yersinia* phages were detected only in samples from milk-fed infants, but not in meconium samples (*p* < 0.05). Additionally, *Aeromonas* phages and *Bacteroidetes* phages were detected only in the breast-fed group, while *Klebsiella* and *Lactobacillus* phages were found only in the formula-fed group.

#### 2.2.3. Virome Differences among the Experimental Groups

The abundance of more than half of *Enterococcus* phages, *Pseudomonas* phages, and *Streptococcus* phages was significantly higher in breast-fed samples (*p* < 0.05), and detected only in a very low proportion of the meconium samples. Additionally, the breast-fed group showed a higher abundance (*p* < 0.05) of *Escherichia* phages than the formula-fed group. In contrast, eleven of thirteen species of CrAss phages and *Staphylococcus* phages SauM Remus were predominant in meconium samples (*p* < 0.05).

Finally, meconium samples presented a higher abundance (*p* < 0.05) of two RNA viruses (*Shamonda* virus and *Oxbow* virus) than the other two groups.

### 2.3. Transkingdom Correlation

Several bacterial families correlate with phages and DNA/RNA viral species. *Pseudomonadaceae* and *Veillonellaceae* families strongly correlated with *Burkholderia* phage phiE094 (0.56 and 0.62, respectively). This makes sense in *Pseudomonadaceae* since phage phiE094 is a lytic phage mainly hosting species from the Proteobacteria phylum, but not in *Veillonellaceae* (Firmicutes phylum). As expected, another positive correlation was observed between a group of *Bacteroidetes* phages and the Bacteroidetes phylum (0.39) (Figure 4). In contrast, the *Serratia* (−0.27), *Stenotrophomonas* (−0.32), *Aeromonas* (−0.31), and *Citrobacter* (−0.30) phages were negatively correlated with the Bacteroidetes phylum. Additionally, as shown in Figure 5, the Bacteroidetes phylum was positively correlated with the *Shamonda* virus (0.47) and *Oxbow* virus (0.50), and negatively correlated with *Enterobacteria* phages (−0.35). Another association found was the positive correlation between *Gammaherpesvirus* (0.3, human virus) and the *Deinococcota phylum*.

### 2.4. SmallRNome Analysis

Starting from an average of 20.4 million single-end reads per sample, an average of 0.26% of the reads were assigned to human small RNA annotations. Collectively, 1918 hsa-miRNAs and 1514 hsa-sncRNAs were assigned with at least one read. For the differential expression analysis, both sets were combined, and only genes with 3 counts in at least 8 samples were considered, resulting in 227 human small RNAs.

In human annotation, in the meconium group, 13% of the assigned reads were for miRNAs, and the remaining 87% were assigned to sncRNA. In the breast-fed group, the percentages were 28% for miRNAs and 72% for sncRNA, and in the formula-fed group, the percentages were 17% for miRNAs and 83% sncRNA. The remaining reads not aligned to hsa-miRNAs and hsa-sncRNAs were further mapped against the human genome to identify those derived from human RNAs. However, 91.12% of the input reads remained unaligned. Then, these reads were mapped to bacterial, archaeal, and viral genomes, receiving the highest percentage of reads with bacteria (on average, 99% of the assigned reads), followed by virus with 0.73%, and less than 0.01% of the aligned reads associated with archaea.

#### 2.4.1. SmallRNA Expression Differences between Meconium and Milk-Fed Sample

Several significant differences were found in small RNA expression when comparing meconium samples with the other two experimental groups (Figure 6). Specifically, ten small RNAs were upregulated, and eight were downregulated in meconium samples compared with formula-fed samples. Similarly, upon comparing meconium with breast-fed samples, twenty small RNAs were upregulated, and ten were downregulated in the meconium group (considering a threshold of 1.5). See Appendix A for further details (Appendix A).

#### 2.4.2. Interference of Meconium miRNA in Metabolic Pathways

For a deeper functional analysis, the target genes for 17 differentially expressed miRNAs were predicted using several online algorithms (described in the methods section) and mapped to the Kyoto Encyclopedia of Genes and Genomes (KEGG) pathways using the KEGG Mapper. False discovery rate (FDR) correction was calculated. As presented in Table 1, six possible pathways were listed with a *p* < 0.001, including the signaling pathways of transforming growth factor β (TGF-β), fatty acid metabolism, and adherens junction, among others.

#### 2.4.3. Main Difference in mascRNA among Milk-Fed Infants

A single significant difference was found between the breast-fed and formula-fed groups: MALAT1-associated small cytoplasmic RNA (ENST00000611300.1) was downregulated in the formula-fed group compared to that in the breast-fed group (considering a threshold of 1.5).

#### 2.4.4. Correlation between small RNA Profile and Bacterial Population

To investigate the relationships between small RNAs and the intestinal microbiome, all small RNA expression levels and microbial family abundance were analyzed together. As shown in Figure 7, the expression of several small RNAs correlated (positively and negatively) with bacterial families. For instance, there were strong positive correlations between MT-IC-201 and five bacterial families: Bacteroidia (0.73), *Peptococcaceae* (0.73), *Spirochaetaceae* (0.60), *Erwiniaceae* (0.62), and *Muribaculaceae* (0.64). mir 4792-201 had a positive correlation with five different families: *Clostridiales* (0.65), *Clostridia* (0.63), *Spirochaetaceae* (0.68), Bacteroidia (0.66), and *Peptococcaceae* (0.60). On the other hand, the Bacteroidales order had the largest number of correlations; in addition to the two previously mentioned correlations, it had six more positive correlations (one mitochondrial RNA (0.62), one SnoRNA (0.60), and four miRNAs (0.6–0.66)) and five negative correlations (mir-103b-1 (−0.58), mir-378a (−0.56), mir-101-1 (−0.56), has-let-7a-2 (−0.58), and mir-103a-2 (−0.57)). The other family that had many correlations was the *Peptococcaceae* family, which presented five positive (two mitochondrial RNAs (0.60) and three miRNAs (0.6–0.64)) and five negative (mir-103b-1 (−0.56), mir-378i (−0.55), mir-101-1 (−0.56), hsa-let-7a-2 (−0.58), and mir-103a-2 (−0.57)) correlations. Finally, the *Muribaculaceae* family had one positive correlation with mitochondrial RNA (0.63).

## 3. Discussion

The gut microbiota, which includes bacteria, archaea, fungi, and viruses, plays an important role in human health. The development of an adult intestinal microbiota begins with the primary colonization of the infant gut, the composition of which may be affected by several early-life factors, such as birth mode or feeding type [25]. In the present study, we investigated the bacteriome and virome composition of neonates at the moment of delivery (meconium), and, we characterized the bacterial and viral populations depending on the feeding type (breast/formula milk) in 2- to 4-month-old infants. As in many other studies, bacteria were detected in all meconium samples [26,27,28], contrasting the past hypothesis that considered meconium sterile [29]. Metagenomic analysis of meconium samples showed large interindividual differences and low species diversity, with Proteobacteria in the highest proportion (42%), followed by Firmicutes (30%), Bacteroidetes (18%), Actinobacteria (10%), Verrucomicrobia (0.4%), and Desulfobacterota (0.2%). These results are consistent with previous studies showing a high abundance of Proteobacteria and a lower abundance of Bacteroidetes [30,31]. The Proteobacteria predomination in meconium samples is explained by its similarity with the bacterial communities found in the mother’s placenta, regardless of the method of delivery, and different from those found in the maternal vagina, according to previous studies [32]. In our study, the most abundant genera in meconium samples were *Escherichia-Shigella*, *Bacteroides*, *Bifidobacterium*, *Streptococcus*, *Clostridia*, *Staphylococcus*, *and Enterococcus*, with the last two previously reported as highly abundant [31,33]. As we expected, meconium samples presented statistically significant differences in the microbial population compared to the 2–4-month-old infant groups. *Muribaculaceae* (*p* < 0.001), *Prevotella* (*p* < 0.01), and the *Propionibacteriaceae* family (*p* < 0.05) were higher in the meconium group than in the infants’ stool from later developmental stages. Despite these differences, the bacterial composition of the meconium group was more similar to that of the formula-fed group than to that of the breast-fed group, suggesting that maternal milk has a higher modulating effect than formula, as published previously [34]. It is worth noting that the virome has not been previously fully explored. In our study, the meconium group presented a high abundance of crAssphages compared to the other two lactating groups, agreeing with the published evidence that crAssphage abundance increases with age [35] and supporting the hypothesis of vertical transmission during delivery from the mother. In contrast, the eukaryotic viruses in the meconium group were in low abundance, and many species were not even present in comparison with the two milk-fed groups. Our results showed that viral species richness presented low values at month 0 (meconium), but higher richness after four months, and reported that the vast majority of taxonomic classifications were phage families, according to a previous study [36].

Regarding the type of feeding, the infant intestinal microbiota was predominantly represented by microorganisms from the Proteobacteria phylum (46% in formula-fed and 28% in breast-fed) and Firmicutes phylum (33% in both groups), followed by those from the Bacteroidetes phylum (11% in the formula-fed group and 23% in the breast-fed group) and Actinobacteria phylum (11% in the formula-fed group and 16% in the breast-fed group). Alpha diversity was higher in the formula-fed group, and beta diversity clearly differentiated the groups into two clusters according to previous studies [33]. In fact, the formula-fed group presented higher levels of *Prevotellaceae* and *Enterobacteriaceae* families and a lower abundance of *Bacteriodaceae*, partially in agreement with a previous study that showed higher levels of *Enterobacteriaceae,* but no changes in *Bacteriodaceae* [37]. Focusing on the virome analysis, there were few significant differences in DNA viruses according to the type of feeding: only *Shigella* phage SfIV, *Burkholderia* phages, and *Streptomyces* phages presented greater relative abundances in the formula-fed group. Additionally, *Escherichia* phage JLK-2012 and *Microviridae* sp., both RNA viruses, were only present in the breast-fed group. These results contrasted with previous studies [18] where viruses infecting human cells were found only in formula-fed infants. Additionally, a previous study [35] showed that after four months of life, human cell viruses were more prominent, including *Adenoviridae*, *Anelloviridae*, *Caliciviridae*, and *Picornaviridae*, but we did not find any of these species; this result can be explained by the size of these viruses that might have remained in the filter during the enrichment steps.

Nevertheless, human pathogens (eukaryotic viruses) are comparatively well documented but are outside of infections, and their abundance is low in a healthy human gut. In contrast, phages are a natural component of every environment’s intricate microbiome, depending on if it is a place where they can live freely, such as the human gut. There is evidence that phages and bacteria coexist and evolve together, but their interaction in the gut environment is poorly described [38]. To analyze this behavior, we correlated both profiles (bacteriome and virome) for the whole cohort. Notably, most of the positive correlations occurred between a bacteriophage and its host, i.e., the greater the presence of a bacteriophage, the greater the presence of its bacterial host. This agrees with the hypothesis that bacteria–phage interactions work as a network in which cross-infective phages invade other bacteria, in addition to their putative bacterial hosts, to obtain a dynamic equilibrium with all microbial communities of the gut microbiome to regulate gut homeostasis [39].

It is important to note that the current knowledge of the virome is very limited, and most previous virome profiling studies have focused on the DNA virome [40]. Here, we carried out two separate protocols for the DNA and RNA viromes, and as expected, we mainly identified both DNA and RNA bacteriophages [40]. However, as in previous studies, several protocol limitations can be identified [41]. First, the experimental enrichment method introduces some bias; for example, large viruses, such as herpesviruses, may be retained in microfiltration. Second, viruses have a small genome, and their proportion in comparison with bacteria and the host genome is very small. Third, two different protocols were needed for the study of RNA and DNA viruses. Virus taxonomy classification is an additional challenge. Whereas viruses are traditionally classified according to their morphology, their classification based on genomic sequences is more complex. Global viral diversity has not yet been characterized, and many viruses do not have reference sequences in databases. Due to this shortcoming, 90% of the virome sequenced reads that we obtained did not share homology with any reference database, a situation also reflected in other published studies [42,43].

While many studies have focused on how crucial a balanced microbiota is for homeostasis, epigenetic signaling controlling microbiota evolution has received less attention. miRNAs have recently been described to interact with the gut microbiota in a reciprocal manner and affect the host’s health status. In this study, we characterized the fecal and, importantly, meconium small RNAs for the first time to determine differences between infant milk feeding and the impact on the intestinal holobiont. To add more evidence to this statement, we integrated bacteriome data with smallRNome to gain a better understanding of the overall regulation. We observed several miRNAs that were expressed significantly differently in meconium compared with both milk-fed group samples. For example, mir-30d and mir-30a were upregulated in the meconium group in comparison with the formula-fed group. These two miRNAs belong to the miR-30 family and play a crucial regulatory role in the development of tissues and organs, and in the pathogenesis of clinical diseases [44]. These miRNAs are, for the first time, linked with infant microbiota or type of feeding. Because these miRNAs come from the mother, it is not surprising to find that newborns may be more protected against developmental problems [45] than milk-fed infants. On the other hand, MALAT1-associated small cytoplasmic RNA was found to be upregulated in breast-fed samples compared to formula-fed samples. This small RNA has been described to regulate TLR-induced proinflammatory and antiviral responses, suggesting its participation in the immune response in the early stages of life [46]. Thus, breast-fed infants show an improved and well-prepared response to inflammation and viral infection. Moreover, mir-24 and mir-29 were downregulated in the meconium group in comparison to in the breast-fed group, and high expression of both miRNAs in maternal plasma was associated with a high risk of preeclampsia [47]. Moreover, mir-21 was downregulated in the meconium group compared to the breast-fed group. Low expression of this miRNA in the placenta was associated with intrauterine growth restriction [48] or with maternal cigarette smoking during pregnancy [49]. Additionally, some metabolic pathways were predicted to be affected by some of these miRNAs that were expressed significantly differently in meconium samples. One of the detected pathways was TGF-β signaling, which regulates many aspects of physiological embryogenesis and adult tissue homeostasis [50], and was associated with infant birth weight [51]. Another metabolic pathway was related to adherens junctions, which may play a crucial role in regulating the intestinal barrier. In addition, we report strong correlations between the *Clostridia*, *Spirochaetaceae*, *Erwiniaceae*, *Peptococcaceae*, RF39, and *Bacteroidia* families and certain miRNAs, such as mir4792-201. This miRNA has been previously described to target the FOXC1 gene (Forkhead Box C1), which plays a role in the oxidative stress response, suggesting that these bacterial taxa are involved in this metabolic process [52]. In addition, the Bacteroidia order had a strong correlation with two mitochondrial genes (ENST00000636729.1 and ENST00000387392.1) and two miRNAs (mir4472-2, mir-10396b), and *Peptoccocaceae* had a negative correlation with mir101-1, which has been described to suppress different virus replications by targeting different genes [53,54]. These findings suggest that the smallRNome mediates host regulation of the intestinal microbiota from early development, which is an effective strategy for establishing a structured and dynamic holobiont. However, for complete understanding, the complex network of interactions between miRNAs and their targets that also depend on the cell type, location, and tissue condition must be considered [55].

Finally, we can surmise that our analysis of the intestinal virome uncovered substantial variation and associations with the corresponding bacteriome and several factors, such as the smallRNome. These results provide the basis for a better understanding of microbial ecology and its relationship with the host.

## 4. Materials and Methods

### 4.1. Study Design and Sample Collection

This study included a total of 64 vaginally delivered healthy full-term infants; 27 of them were newborns, and meconium samples were obtained within 0–48 h after birth; the remaining 37 were infants under four months old (3.2 ± 0.7) and were divided into 25 breast-fed and 12 formula-fed infants. To ensure the accuracy of our results, we excluded infants whose mothers had illnesses or were on medication during pregnancy, as well as those who had undergone restricted diets. Newborns with pathologies were also excluded, as were infants whose birth weight fell below the 25th or above the 75th Spanish percentile. The breast-fed group was formed by infants exclusively fed breast milk for 2–4 months after birth. The formula-fed group was formed by infants exclusively fed formula for 2–4 months, whose mothers voluntarily chose to feed their babies formula. The formula milk used in this study was compliant with the Commission Delegated Regulation (EU) 2016/127 regarding composition. While different commercially available formula milks were used, they all met the same nutritional requirements. For all cases, stool samples were collected and immediately frozen at −20 °C. Ethical approval was granted on 28 June 2018. Informed consent was obtained from the parents of the eligible infants. This study is in accordance with the ethical standards of the Declaration of Helsinki.

### 4.2. Bacteriome Analysis

Bacterial DNA was extracted from approximately 200 mg of meconium or fecal sample using a QIAamp^®^ DNA Stool Mini Kit (Qiagen Inc., Hilden, Germany) according to the manufacturer’s instructions. DNA quantity and purity were assessed using a NanoDrop 2000 spectrophotometer (Thermo Fisher Scientific, Waltham, MA, USA).

The V4 variable region of the 16S rRNA gene was amplified by PCR as described previously [56].

The amplicon libraries were pooled and diluted to 35 pM before clonal amplification. The Ion 510 and 520 and 530 Ion Chef Kit (Life Technologies, Carlsbad, CA, USA) was employed for template preparation. Next-generation sequencing of the clonally amplified 16S rRNA libraries was performed on an Ion GeneStudio S5 system (Life Technologies, Carlsbad, CA, USA) following the manufacturer’s instructions. The generated reads were quality filtered, analyzed with QIIME2 (2022.2), and passed for classification into amplicon sequence variants (ASVs) to DADA2, using only reads of at least 200 bp and truncating at that length. These ASVs were taxonomically classified with VSEARCH against the Silva database at 99% homology [56]. Alpha diversity was calculated as Shannon, Chao1, and Faith indices, and beta diversity was calculated with Bray-Curtis, Aitchison, Jaccard, and UniFrac (weighted and unweighted) distances. PERMANOVA and ANOSIM analyses were performed on the beta diversity data. The taxonomic abundance for each taxon at every level was compared using a Kruskal-Wallis test and corrected by Holm-Sidak correction. Statistical power analysis for the comparisons was calculated according to Equations D and E from Ferdous et al. [57]. The results are shown in Appendix A.

### 4.3. Virome Analysis

Each sample was weighed and resuspended to a final concentration of 10% (*w*/*v*) in autoclaved phosphate-buffer saline (PBS) buffer (Thermo Fisher Scientific, Waltham, MA, USA) and vigorously vortexed until reaching a completely homogeneous suspension. The suspension was centrifuged at 4800× *g* for 10 min at 4 °C to remove/clarify large particles that may be present in the samples, such as organic matter or host cells. The supernatant was collected and filtered through a 0.22 µm filter Steritop (Millipore Sigma, Hayward, CA, USA) to retain the bacterial cells. The filtrate was ultracentrifuged (14 mL, Polypropylene Tube, 14 × 95 mm—50 Pk, and SW 40 Ti Swinging-Bucket Rotor Package) at 900,000× *g* for 90 min at 4 °C to reduce the liquid volume and concentrate VLPs. The pellet was resuspended in 199 µL of enzyme buffer and treated with 25 U of Benzonase^®^ Nuclease (Millipore Sigma, Hayward, CA, USA) at 37 °C for 90 min to digest non-particle-protected nucleic acids. Subsequently, viral nucleic acids were extracted by a Quick-DNA/RNA Viral Kit (Zymo Research, Irvine, CA, USA) according to the manufacturer’s instructions and eluted into 36 µL of RNase-free water. The DNA and RNA purity were evaluated by the A260/A280 ratio using a NanoDrop 2000 spectrophotometer (Thermo Fisher Scientific, Waltham, MA, USA). Nucleic acids were divided into two aliquots, one of which was treated with 1 U of DNase I (Invitrogen™, Waltham, MA, USA) at 37 °C for 90 min to obtain pure RNA, and the other aliquot was treated with 10 U of RNase ONE™ ribonuclease (Promega, Madison, WI, USA) at 37 °C for 30 min to obtain pure DNA.

Afterwards, DNA libraries were generated by the Ion Xpress™ Plus Fragment Library Kit for the AB Library Builder™ System (Thermo Fisher Scientific, Waltham, MA, USA) according to the manufacturer’s instructions, and libraries were reamplified and purified manually after their creation. RNA libraries were created by a Total RNA-Seq Kit v2 (Thermo Fisher Scientific, Waltham, MA, USA) for whole transcriptome libraries according to the manufacturer’s instructions. DNA and RNA libraries were quantified by electrophoresis at TapeStation using High-sensitivity DNA ScreenTape Analysis (Agilent Technologies, Santa Clara, CA, USA). DNA and RNA template libraries were performed using the Ion Chef System (Thermo Fisher Scientific, Waltham, MA, USA) and sequenced using the Ion GeneStudio S5 System (Thermo Fisher Scientific, Waltham, MA, USA). All steps in the Ion GeneStudio S5 System (Life Technologies, Carlsbad, CA, USA), including amplification through sequencing, were performed according to the manufacturer’s recommendations.

For both DNA and RNA, each sample’s reads were assembled using SPAdes genome assembler 3.15.5. The resulting contigs were classified as viral, potentially viral, or nonviral using viralVerify 1.1 (https://github.com/ablab/viralVerify, accessed on 9 September 2022).

With the Pfam-A HMM database and a BLAST against the NCBI “nt” database, contigs marked as nonviral or that matched against cellular organisms or plasmids were discarded. The abundance of each contig within each sample was derived from its coverage. The remaining contigs were assigned a taxonomy based on their best match in the BLAST results. The contigs from all samples were clustered at 90% homology to form OTUs (operational taxonomic units) comparable across samples. From these OTUs, alpha and beta diversity analyses and statistical analyses, such as those for the bacteriome, were performed, with the only difference being avoiding metrics that rely on phylogenetic distances. The results are shown in Appendix A.

### 4.4. Analysis

Total RNA was extracted from 250 mg of meconium or fecal samples using Direct-zol™ RNA Miniprep Plus (Zymo Research, Irvine, CA, USA) according to the manufacturer’s instructions, and the RNA concentration was quantified using a NanoDrop 2000 spectrophotometer (Thermo Fisher Scientific, Waltham, MA, USA). Libraries were created by a TruSeq Small RNA Library kit (Illumina, San Diego, CA, USA) according to the manufacturer’s instructions, and were sequenced by Illumina NextSeq2000. sRNA-seq pipeline analysis was performed following a previously described approach [58,59,60].

FastQC software v0.11.9 (http://www.bioinformatics.babraham.ac.uk/projects/fastqc/) was used for quality control (QC) of FASTQ files. Samples were preprocessed with Cutadapt (version 2.9), discarding reads shorter than 14 nt and imposing a maximum error rate equal to 0.15 for mismatches, insertions, and deletions. Trimmed reads were mapped against hsa-miRNAs from miRBase using the BWA Algorithm v. 0.7.17-r1188. Unaligned reads were aligned against hsa-sncRNA sequences shorter than 80 bp from Ensembl with BWA default parameters. The quantification of miRNA and sncRNA was performed with SAMtools and merged into a unique smallRNA count matrix. From this count matrix, differential expression analysis was performed with DESeq2.

The reads that were left unmapped were aligned with BWA against the Hg38 genome from Ensembl. The reads that were still unmapped on the human genome were then analyzed by Kraken for metatranscriptomic analysis. The statistical power of the comparisons for each miRNA was calculated according to Equations D and E from Ferdous et al. [57]. The results are shown in Appendix A.

### 4.5. Metabolic Pathway Analysis

Target genes for 14 miRNAs that were significant, when comparing meconium with the other two groups, were predicted using at least four of these public database algorithms online: Diana MicroT, miRanda, miRDB, PicTar, and miRNAMap. Target genes predicted by at least two different tools were mapped to the Kyoto Encyclopedia of Genes and Genomes (KEGG) pathways using KEGG Mapper, and enriched by Fisher’s exact test (confidence interval 95%) with FDR correction using R Software version 4.2.2 (R Development Core Team, 2013, Vienna, Austria).

### 4.6. Omics Data Integration

Integration of the bacteriome, RNA and DNA virome, and sRNA transcriptome was performed in R version 4.2.2 with the mixOmics package [61]. We evaluated each pair of omics data because of sample limitations. We calculated the sPLS model to identify the most discriminative features between each pair of omics data to evaluate their associations. sPLS in regression mode was applied to rlog-normalized sncRNA counts and log-ratio-transformed relative abundance for the bacteriome and virome, respectively. The models were tuned based on 10-fold cross-validation, and optimal parameters were chosen according to the highest mean correlation measure for each pair of omics.

## 5. Conclusions

Despite the limited number of samples available from the formula-fed group, our results indicate that the gut environment of newborns, as assessed through three examined omic levels, is more similar to that of the formula-fed group than to the breast-fed group. This supports the idea that diet has a significant impact on gut microbiota, and confirms the modulating effects of breast-feeding, likely mainly triggered by the intake of colostrum.

Furthermore, our study highlights a clear difference in the virome between newborns and four-month-old infants, underscoring the dynamic nature of viral populations and their role in shaping community assembly and host health. Notably, we found transkingdom correlations between virome components and bacteria, suggesting additional layers of complexity in host–microbial homeostasis. While we acknowledge certain protocol limitations and analysis shortcomings, our findings point to the need for further investigation into these areas.

In addition, our study is the first to identify miRNAs in meconium samples, indicating the existence of epigenetic mechanisms before birth that putatively interact with the host intestinal system, modulating and controlling its homeostasis. We also found that the miRNome profile, similar to the virome, rapidly changes in the first months of life. However, additional research is needed to fully understand the precise mechanisms underlying the interaction between the microbiota and miRNA.

In conclusion, our findings suggest that the gut environment is rapidly acquired after birth and is highly malleable, with environmental factors and genetic responses creating complex molecular interactions at the host–microbiota interface. These interactions may play a significant role in regulating predisposition to future diseases.

## Figures and Tables

**Figure 1 ijms-24-08069-f001:**
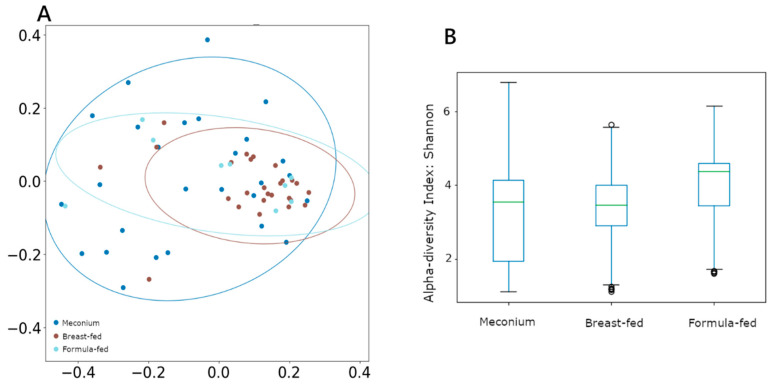
(**A**) β-diversity measured using unweighted UniFrac distances. (**B**) α-diversity measured using Shannon, with the Y-axis representing the Shannon score values.

**Figure 2 ijms-24-08069-f002:**
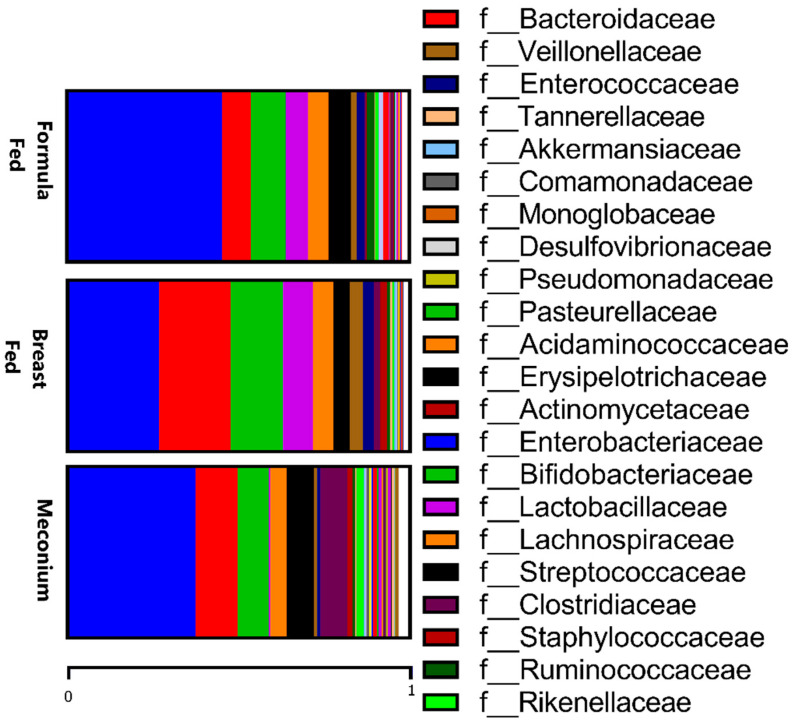
Relative abundances of taxa at the family level in the three studied groups.

**Figure 3 ijms-24-08069-f003:**
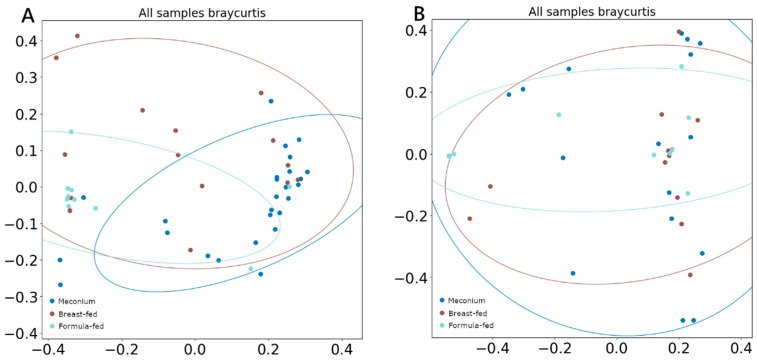
PCoA (**A**) β-diversity of DNA viruses measured with Bray–Curtis, (**B**) β-diversity of RNA viruses measured with Bray–Curtis.

**Figure 4 ijms-24-08069-f004:**
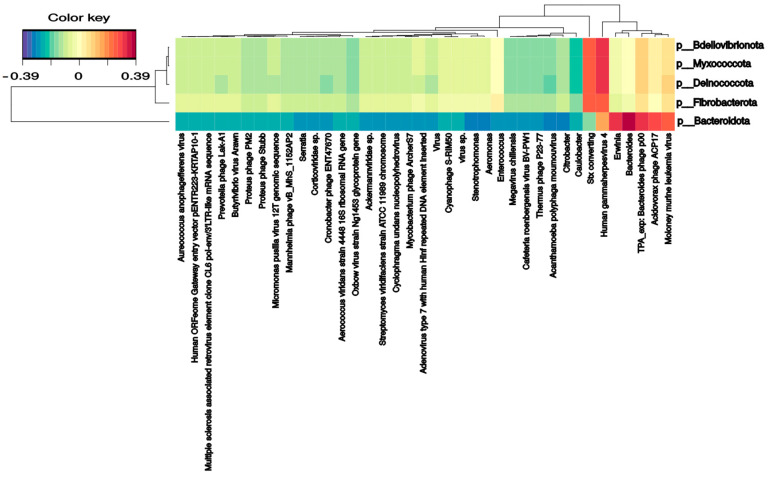
Clustered image map from the sparse partial least squares (sPLS) on the DNA virus and phages (on the X-axis) and bacterial phyla (on the Y-axis). It shows the relationship between the variables of each omic dataset. A ± 0.25 threshold was used.

**Figure 5 ijms-24-08069-f005:**
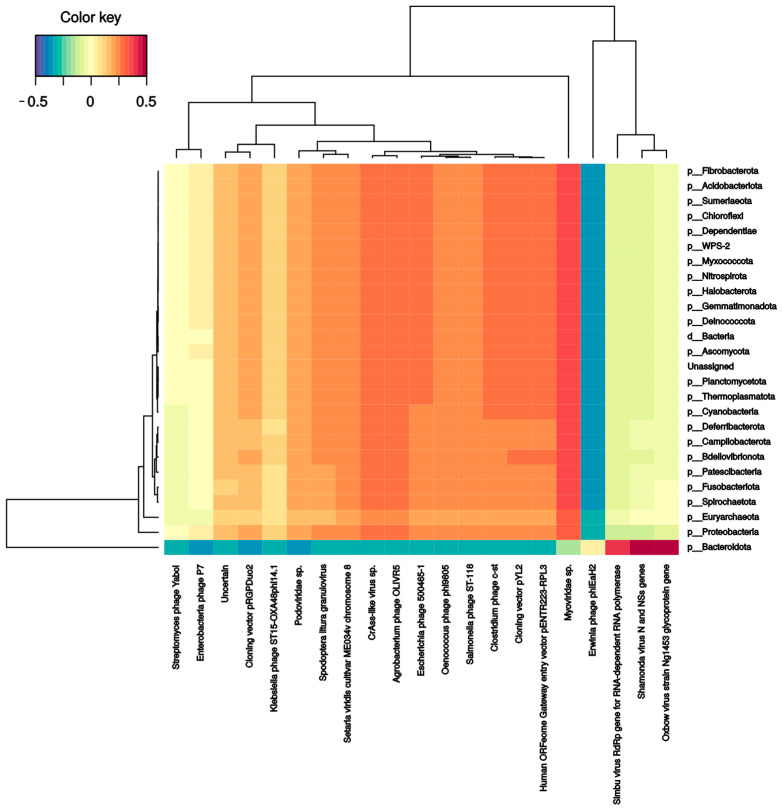
Clustered image map from the sparse partial least squares (sPLS) on the RNA virus (on the X-axis) and bacterial phyla (on the Y-axis). It shows the relationship between the variables of each omic dataset. A ± 0.3 threshold was used.

**Figure 6 ijms-24-08069-f006:**
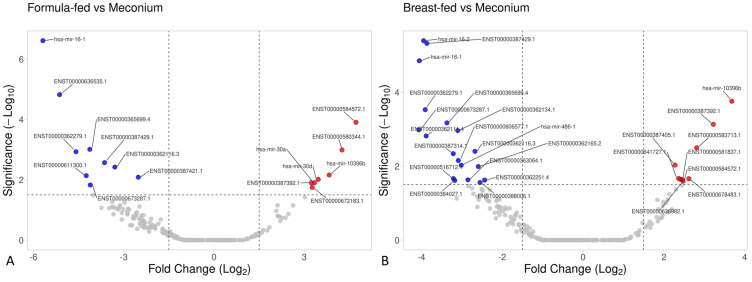
Volcano plot based on small RNA data. (**A**) The meconium group versus the breast-fed group, and (**B**) the meconium group versus the formula-fed group.

**Figure 7 ijms-24-08069-f007:**
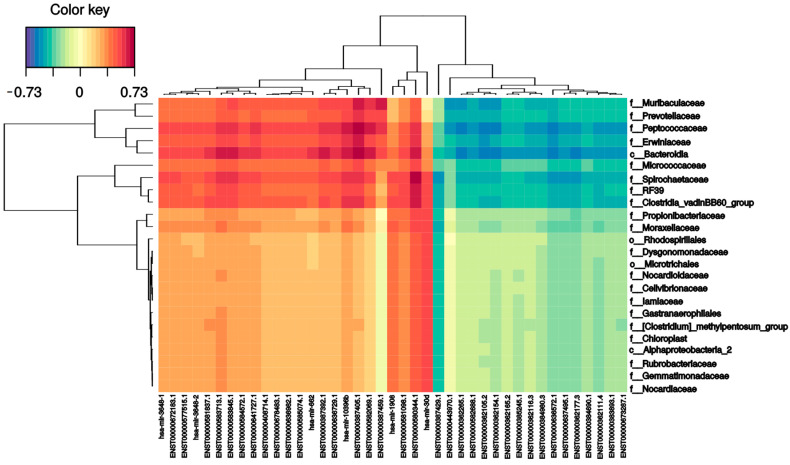
Sparse partial least squares (sPLS). The following graph indicates the correlation between the variables of each dataset: X-axis with sncRNAs, and the Y-axis with the bacteriome highest taxonomic level that can be classified (p_phylum, c_class, o_order, f_family). The ±0.5 threshold was used for generation of the figure.

**Table 1 ijms-24-08069-t001:** Significantly targeted pathways.

KEGG Pathway	FDR Adjustment	Targeted Genes	Number of Involved miRNAs
Fatty acid biosynthesis	<0.001	7	6
Cell cycle	<0.001	87	6
Fatty acid metabolism	<0.001	31	7
Adherens junction	<0.001	55	9
Lysine degradation	<0.001	33	9
TGF-beta signaling pathway	<0.001	55	7

## Data Availability

The data that support the findings of this study are openly available in BioProject at [http://www.ncbi.nlm.nih.gov/bioproject/949443, accessed on 28 April 2022], reference number BioProject ID: PRJNA949443, except data coming from smallRNA analysis, since consent was not obtained from participants.

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
