# Peer review of "Gut Microbiome and Small RNA Integrative-Omic Perspective of Meconium and Milk-FED Infant Stool Samples"

_ijms, 2023, doi:10.3390/ijms24098069_

Round 1

Reviewer 1 Report

Regarding the manuscript entitled “Bacteriome, virome and smallRNome integrative-omics analysis to decipher host-microbiome interaction in meconium from new-born and stool from milk-fed infants”, the authors aimed to characterize the bacteriome, virome, smallRNome and its interaction in meconium and stool samples from infants.

The study is interesting. However, a revision is needed before its acceptance.

The title must be revised and improved. As a suggestion “characterization of the bacteriome, virome, smallRNome and its interaction in meconium and stool samples from infants fed on breast milk or formula milk”

Cohort description must be transferred to experimental design in the material and methods section.

Please write the details of “formula-fed infants”. Was it one formula?

More details regarding the weight of infants in the three groups must be added. Additionally, more details regarding the dams of infants in the three groups must be added. The health status and the nutrition behavior of the dam can affect the results.  Did these infants or dams receive any medication?

The conclusion section must be summarized. Please avoid the re-description of your methodology in the conclusion section.  

Author Response

We would like to express our gratitude to the reviewer for the valuable comments and suggestions. Based on the feedback, we have made the requested changes to the manuscript to improve its clarity and quality. The revisions we made are as follows:

  • We agree with the reviewer's suggestion to clarify the title and have accordingly revised it. The new title provides a clearer indication of the focus of our study.
  • In response to the issues raised by the reviewer regarding the cohort description, we have taken several steps to improve this section. Specifically, we transferred the cohort description to the Materials and Methods section for greater clarity and added more details regarding our inclusion and exclusion criteria, with a particular focus on dams and formula milk characteristics.
  • We have also addressed the reviewer's concerns regarding the conclusion section by rephrasing certain sentences to avoid unnecessary repetition.

We believe that the revisions we made have significantly improved the clarity of the manuscript.

Reviewer 2 Report

The article entitled Bacteriome, virome and smallRNome integrative-omics analysis to decipher host-microbiome interaction in meconium from new-born and stool from milk-fed infants by Polina et al is based on a very good idea. The authors tried to enlighten and To gain a better understanding of how the gut microbial composition is shaped in early life stages, we conducted a fecal holo-omic study in newborn infants (meconium samples were obtained) and in exclusively milk-fed infants either with human milk  (breast-fed group) or formula milk (formula-fed group). This study included the analysis  of fecal bacterial and viral population as well as the identification of host small RNA signali To gain a better understanding of how the gut microbial composition is shaped in 84 early life stages, we conducted a fecal holo-omic study in newborn infants (meconium samples were obtained) and in exclusively milk-fed infants either with human milk (breast-fed group) or formula milk (formula-fed group). This study included the analysis of fecal bacterial and viral population as well as the identification of host small RNA signaling in feces.

The article can be accepted after minor revision and answering my comments.

My comments are

Introduction:

Introduction needs improvement and latest articles should be cited in it.

The authors should cite the following articles

https://www.frontiersin.org/articles/10.3389/fmicb.2023.1157615/abstract

https://www.mdpi.com/1420-3049/28/2/491

doi: 10.3390/molecules27175399

Materials and methods:

What was the specific reason of selecting (24-48 hours ) new born babies?

Formula fed babies (Their was problem in breast feeding or the moms voluntarily agreed to fed their babies of formula).

As we know that Colostrum is the milk produced during the first few days after birth and contains high levels of immunoglobulins, antimicrobial peptides, and growth factors. Colostrum is important for supporting the growth, development, and immunologic defence of neonates. Why the authors didn’t took this into account by comparing the breast fed babies microbiota to formula fed babies microbiota?

Results and discussions are well explained.

Conclusion is very well written.

There are some topographical mistakes which should be removed.

English language correction is required.

The article entitled Bacteriome, virome and smallRNome integrative-omics analysis to decipher host-microbiome interaction in meconium from new-born and stool from milk-fed infants by Polina et al is based on a very good idea. The authors tried to enlighten and To gain a better understanding of how the gut microbial composition is shaped in early life stages, we conducted a fecal holo-omic study in newborn infants (meconium samples were obtained) and in exclusively milk-fed infants either with human milk  (breast-fed group) or formula milk (formula-fed group). This study included the analysis  of fecal bacterial and viral population as well as the identification of host small RNA signali To gain a better understanding of how the gut microbial composition is shaped in 84 early life stages, we conducted a fecal holo-omic study in newborn infants (meconium samples were obtained) and in exclusively milk-fed infants either with human milk (breast-fed group) or formula milk (formula-fed group). This study included the analysis of fecal bacterial and viral population as well as the identification of host small RNA signaling in feces.

The article can be accepted after minor revision and answering my comments.

My comments are

Introduction:

Introduction needs improvement and latest articles should be cited in it.

The authors should cite the following articles

https://www.frontiersin.org/articles/10.3389/fmicb.2023.1157615/abstract

https://www.mdpi.com/1420-3049/28/2/491

doi: 10.3390/molecules27175399

Materials and methods:

What was the specific reason of selecting (24-48 hours ) new born babies?

Formula fed babies (Their was problem in breast feeding or the moms voluntarily agreed to fed their babies of formula).

As we know that Colostrum is the milk produced during the first few days after birth and contains high levels of immunoglobulins, antimicrobial peptides, and growth factors. Colostrum is important for supporting the growth, development, and immunologic defence of neonates. Why the authors didn’t took this into account by comparing the breast fed babies microbiota to formula fed babies microbiota?

Results and discussions are well explained.

Conclusion is very well written.

There are some topographical mistakes which should be removed.

English language correction is required.

Author Response

We would like to extend our appreciation to the reviewer for their valuable feedback and suggestions. In response to the comments, we have incorporated the suggested changes into the manuscript to enhance its clarity and quality. The revisions we have made are outlined below:

  • The introduction has been updated, and several recent references have been added. In particular, the following articles have been cited:
  • Xiao L, Zhao F. Microbial transmission, colonisation and succession: from pregnancy to infancy. Gut. 2023 Apr;72(4):772-786. doi: 10.1136/gutjnl-2022-328970. Epub 2023 Jan 31. PMID: 36720630; PMCID: PMC10086306.
  • Fulci V, Stronati L, Cucchiara S, Laudadio I, Carissimi C. Emerging Roles of Gut Virome in Pediatric Diseases. Int J Mol Sci. 2021 Apr 16;22(8):4127. doi: 10.3390/ijms22084127. PMID: 33923593; PMCID: PMC8073368.
  • Nishijima S, Nagata N, Kiguchi Y, Kojima Y, Miyoshi-Akiyama T, Kimura M, Ohsugi M, Ueki K, Oka S, Mizokami M, Itoi T, Kawai T, Uemura N, Hattori M. Extensive gut virome variation and its associations with host and environmental factors in a population-level cohort. Nat Commun. 2022 Sep 6;13(1):5252. doi: 10.1038/s41467-022-32832-w. PMID: 36068216; PMCID: PMC9448778.

  • We have addressed the issues raised regarding the Materials and Methods section and provided more details regarding the inclusion and exclusion criteria, with a particular focus on dams and formula milk characteristics.
    • In particular, we added the following: “The formula-fed group was formed by infants fed formula exclusively for 2-4 months, whose mothers voluntarily chose to feed their babies formula. The formula milk used in this study was compliant with the Commission Delegated Regulation (EU) 2016/127 regarding composition”
    • For newborn babies, samples were collected within 48 h after birth, during which time meconium (the first stool of a newborn) was We have amended this sentence in the manuscript by adding “[…] newborns and meconium samples were obtained within 0-48 h after birth”.
  • We appreciate the reviewer's comments regarding the importance of colostrum in establishing a healthy microbiome. To address this concern, we added a sentence in the conclusion section emphasizing the crucial role of colostrum in newborn health and homeostasis.
  • In addition, the English language has been corrected by American Journal Experts.

We hope that these revisions have improved the manuscript and addressed the concerns raised by the reviewer.

Reviewer 3 Report

Although the topic discussed in this paper entitled:" Bacteriome, virome and smallRNome integrative-omics analysis to decipher host-microbiome interaction in meconium from new-born and stool from milk-fed infants." is of interest to readers, and scientifically relevant, some major revisions are needed before publication.

Major concern: As stated by the authors in the conclusions (line 510), due to the small number of samples, a power analysis for determining statistical power, and effect size is required.

Here follow the other revision listed according to page lines :

Introduction

Line 35: Even if correctly cited in the Discussion section, Archea is missing in the Introduction. Here are some recent references for your knowledge from PubMed, feel free to use others:

Kim, J.Y., Whon, T.W., Lim, M.Y. et al. The human gut archaeome: identification of diverse haloarchaea in Korean subjects. Microbiome 8, 114 (2020). https://doi.org/10.1186/s40168-020-00894-x

Hoegenauer, C., Hammer, H.F., Mahnert, A. et al. Methanogenic archaea in the human gastrointestinal tract. Nat Rev Gastroenterol Hepatol 19, 805–813 (2022). https://doi.org/10.1038/s41575-022-00673-z

Line 35-38. “gut microbiota” is repeated several times in the sentence, please rephrase and/or find synonyms

Line 45: full stop missing after references 7,8.

Line 59: Reference missing

Line 63: replace “named” with namely and add a reference.

Line 64-65 “intestinal” is repeated several times in the sentence, please rephrase and/or find synonyms

Line 69: “and like in the bacteriome…” remove “in the”, “and like bacteriome”

Results

Replace all sub-section titles (from 2.2.1 to the end) with a more concise and impactful title to facilitate reading

Discussion

Line 280: “….. samples. showed large interindividual…” remove full stop.

The manuscript is well written, the few errors found were listed in the comments

Author Response

We greatly appreciate the thoughtful and insightful comments of the reviewer, which have greatly enhanced the rigor and impact of our paper. We have carefully considered the suggestions and have made the following revisions to the manuscript:

  • The introduction has been updated with recent references, including those suggested by the reviewer.
  • We corrected typographical errors throughout the text and rephrased some sentences to eliminate redundancy, as suggested by the reviewer.
  • The subsection titles have been rewritten to convey the content more effectively.
  • We conducted a statistical power analysis and added results to the Materials and Methods section to more clearly describe the study's findings.

We are confident that the changes we have implemented greatly enhanced the clarity of the manuscript.

Round 2

Reviewer 1 Report

The manuscript was improved and can be accepted in its current form.

Reviewer 3 Report

This reviewer is satisfied with the changes made by the authors